# The Use of Galactomannan Antigen Assays for the Diagnosis of Invasive Pulmonary Aspergillosis in the Hematological Patient: A Systematic Review and Meta-Analysis

**DOI:** 10.3390/jof9060674

**Published:** 2023-06-15

**Authors:** Lydia M. P. Bukkems, Laura van Dommelen, Marta Regis, Edwin van den Heuvel, Laurens Nieuwenhuizen

**Affiliations:** 1Maxima Medical Centre, De Run 46000, 5504 DB Veldhoven, The Netherlands; laurens.nieuwenhuizen@mmc.nl; 2Ziekenhuis Gelderse Vallei, Willy Brandtlaan 10, 6716 RP Ede, The Netherlands; ldommelen@zgv.nl; 3Mathematics and Computer Science, Eindhoven University of Technology, Building Helix, Postbus 513, 5600 MB Eindhoven, The Netherlands; marta.regis@mmc.nl (M.R.); e.r.v.d.heuvel@tue.nl (E.v.d.H.)

**Keywords:** invasive pulmonary aspergillosis, galactomannan, serum, broncho-alveolar lavage, hematology, malignancy, systematic review, meta-analysis

## Abstract

The optimal cut-off value of the optical density index of the galactomannan antigen assays (GM) for diagnosing invasive pulmonary aspergillosis in hematological patients is a disputed topic. This article conducts a systematic review with a meta-analysis to establish which optical density index (ODI) cut-off value should be implemented into clinical practice. Pubmed, Embase and Cochrane databases were searched (N = 27). The pooled data, using a generalized linear mixed model with binomial distribution, resulted in an overall serum sensitivity of 0.76 and a specificity of 0.92. For serum ODI 0.5 there was a pooled sensitivity of 0.92 and a specificity of 0.84. The pooled data of all broncho-alveolar lavage (BAL) studies resulted in an overall sensitivity of 0.80 and a specificity of 0.95. For BAL ODI 0.5, there was a pooled sensitivity of 0.75 and a specificity of 0.88. For the BAL ODI 1.0 pooling, the studies resulted in a sensitivity of 0.75 and a specificity of 0.96. Serum ODI of 0.5 and BAL ODI of 1.0 are the most suitable cut-offs for clinical practice. However, our study affirms that the evidence for the use of GM in clinical practice for the hematological malignancy patient is currently insufficient and more research is needed to determine the diagnostic value of GM.

## 1. Introduction

Patients with hematological malignancies are often treated with chemotherapy; this, or the disease itself, makes them immunocompromised, which greatly increases the risk for opportunistic infections such as invasive pulmonary aspergillosis (IPA). IPA is a pulmonary fungal infection caused by the Aspergillus species, most commonly the *A. fumigatus*, *A. flavus*, *A. niger* and *A. terreus* [1]. IPA is a major cause of mortality in this patient group [2]. ICU admission and 90-day mortality rates of 58.4% and 75.2%, respectively, have recently been reported [3]. Establishing an early diagnosis and subsequently starting early treatment has improved the survival rates in these patients [4]. However, due to the unspecific clinical and radiological features combined with the often-poor yield of cultures, diagnosing IPA remains difficult.

To facilitate the diagnosis and treatment of IPA, the European Organization for Research and Treatment of Cancer/Mycosis Study Group (EORCT/MSG) has defined three categories: proven, probable and possible aspergillosis. These EORTC/MSG criteria have recently been revised and updated, leading to a change in the diagnostic criteria, especially for the probable category. One of these changes applies to the detection of the aspergillus-specific marker, galactomannan (GM), in serum and broncho-alveolar lavage (BAL) [5]. GM is a polysaccharide on the cell wall of *Aspergillus* spp., which can be detected with an enzyme-linked immunosorbent assay (ELISA). This assay provides the optical density index (ODI), which can be translated into clinical practice [6]. The appropriate cut-off for this index, however, is a subject of debate. Originally, the serum cut-off for positivity was set at ≥1.5; however, the 2008 EORCT/MSG guideline advised to lower it to ≥0.5 after a review by the FDA [7]. Similarly, in the most recent Dutch Fungal guideline (2017), which is based on the EORCT/MSG report of 2008, a serum galactomannan cut-off value of ≥0.5 and a BAL cut-off value of ≥0.8 were recommended [8]. However, in the updated EORCT/MSG report of 2020, the cut-off was increased to a cut-off value of ≥1.0 for both or when combined lower cut-offs, ≥0.7 for serum and ≥0.8 for BAL, are advised [5]. 

There have been several reviews regarding this topic. However, most reviews are outdated and have not specified the use in the adult hematological patient population, nor have the authors combined both serum and BAL GM in this group [9,10,11]. Since, on the one hand, IPA has a high mortality, but on the other hand, overtreatment should be averted due to potential medication toxicity and induction of antifungal resistance and costs, there is a considerable need for a well-effectuated meta-analysis considering GM to optimize IPA diagnosis. In this systematic review, we want to evaluate the most recent evidence regarding the ODI for serum and BAL ELISA GM for the diagnosis of IPA, which is a key element in the diagnosis of IPA in this patient group.

## 2. Materials and Methods

This systematic review was written according to the PRISMA (Preferred Reporting Items for Systematic Reviews and Meta-Analysis) checklist [12]. 

### 2.1. Eligibility Criteria

Articles were eligible for inclusion if they reported the use of galactomannan Platelia ELISA, in serum or BAL, for the diagnosis of invasive pulmonary aspergillosis in the adult (18+) intensive hematological malignancy population. Any study design, except for a review, was included for analysis. Only articles using the EORTC/MSG criteria to define the IPA cases and controls in proven/probable/possible and no-IPA, were included.

Articles were excluded when the full text was not available, thus excluding all conference abstracts, when the language was not English or Dutch or when the patient population had other significant diseases such as COVID-19, an influenza virus infection, HIV/AIDS and/or all other significant bacterial or viral co-infections. Furthermore, publications where the entire population was currently using posaconazole prophylaxis and publications including the intensive care unit (ICU) population were excluded. All results had to be presented in a manner by which a 2 × 2 table could be formed to analyze the sensitivity and specificity of the GM assay. The data also had to be available in a manner by which the possible IPA patients could be separated from the no-IPA patients. If not, the article was excluded.

### 2.2. Search Strategy

An electronic search in Pubmed, Embase and Cochrane was performed on 13 April 2021. The complete search strategy can be viewed in Appendix A. No limitations or filters were used.

### 2.3. Selection Process

Two independent reviewers (L.B. and L.N.) assessed all articles retrieved through the search. First, all the duplicates were removed with the Wichor method in EndNote [13]. For the article selection, Rayyan was used [14]. After uploading all articles in Rayyan, they were assessed for eligibility on title and abstract using the in- and exclusion criteria. Of all remaining articles of which title and abstract information was insufficient to assess the article, if possible, full text articles were retrieved. Articles that did not have full text availability were excluded. The in- and exclusion criteria were used to make the final selection of articles for inclusion in this review. Any disagreements between the two reviewers L.B. and L.N. were resolved through discussion or, when necessary, a third reviewer (L.v.D.) was consulted.

### 2.4. Data Collection Process

Every included article was thoroughly analyzed and the available data were collected by one reviewer (L.B.). The following data, when available, were collected and recorded in Excel: author, year of publication, country, means of data collection, specific patient population, antifungal prophylaxis, sampling method, galactomannan assay, sample size, age (mean/range), reference test, ODI cut-off’s for positivity, serum/BAL OD index, median GM index, number of positive samples, total samples, total incidence/prevalence, mortality, fungal culture, HRCT, histopathology, number of proven/probable/possible and no-IPA patients and the data for a 2 × 2 table. All data extracted from the article were discussed and reviewed by a second reviewer (L.N.). No study investigators were contacted for additional data.

### 2.5. Methodological Quality Assessment

The methodological quality of all publications included in the meta-analysis was graded using the QUADAS-2 tool. Using this tool, the articles were independently assessed by two reviewers (L.B. and L.N.). After individual assessment, the conflicts were reviewed and resolved through discussion. The QUADAS-2 tool consists of four domains, and for each domain the risk of bias was considered to be unclear, low or high. These risks were estimated based on answering questions about the methodological quality. In three domains, the applicability of the studies was also examined, resulting in grading any concerns about the applicability to be unclear, low or high.

Since galactomannan is incorporated in the EORTC/MSG criteria, there is a risk of incorporation bias in studies which include the galactomannan results to define the reference standard. Thus, this additional criterium was added to the bias assessment in domain three, namely the reference test.

### 2.6. Outcome Measures

The primary outcome is the pooled sensitivity and specificity of serum and BAL galactomannan, and secondarily to compare the sensitivity and specificity per OD index. The objective is to compare different cut-off values for the OD index and decide which cut-off is most adequate to diagnose IPA. Additionally, to confirm which cut-off should be incorporated in clinical practice while also considering the different IPA classifications, it is necessary to compare proven, probable and possible IPA with not having IPA.

### 2.7. Statistical Analysis and Data Synthesis

The EORTC/MSG criteria were used as a reference standard, thus defining four patient groups: proven, probable, possible and no-IPA [7]. In the primary analysis, we defined the proven and probable patients as having IPA and compared them to the no-IPA group as negative controls. For a second analysis, we also presented data including the possible patient group in the IPA group and thus compared proven, probable and possible IPA to the no-IPA controls.

With these classifications, 2 × 2 tables could be constructed; IPA and no-IPA versus a positive or negative galactomannan result. This allowed us to calculate the test accuracy of GM. 

A meta-analysis of the individual study results was performed to obtain the overall sensitivity and specificity across studies.

Our primary goal was to assess the pooled diagnostic power and secondarily to assess the diagnostic power of each cut-off. For both analyses, we fitted a generalized linear mixed model with a binomial distribution for the counts of positive tests, with two (possibly correlated) random effects for the derived sensitivity and specificity. This model has proven to be equivalent to the hierarchical summary ROC model [14,15,16]. Furthermore, this model could be adapted to test the significance of the difference in diagnostic accuracy among specific settings of interest. In particular, for serum GM ODI 0.5, some studies used the cut-off for a single sample for positivity and others used consecutive samples for positivity. To analyze whether this has any effect on the diagnostic accuracy, we compared the two approaches. 

Since we were also interested in the value of serial serum GM screening in the asymptomatic population, we performed a sub-analysis differentiating a population with a higher pre-test probability for IPA, i.e., the studies with populations including patients with fever.

Individual study results were plotted in forest plots together with the pooled values, to give a visual representation of the results of the meta-analysis. All analyses were performed in SAS 9.4, version 9.4.

## 3. Results

### 3.1. Article Selection

The full selection process can be reviewed in Appendix A. A total of 27 articles [2,17,18,19,20,21,22,23,24,25,26,27,28,29,30,31,32,33,34,35,36,37,38,39] were included in the meta-analysis: 15 for the serum GM and 12 for the BAL GM. 

### 3.2. Study Characteristics

#### 3.2.1. Serum

In Appendix A, all study characteristics of the included studies about serum galactomannan are presented. In total, 15 studies were included with a sample size of 2568 patients including 501 patients with proven or probable IPA and 300 patients with possible IPA. All study populations comprised adult (18+) patients with a hematological malignancy. The EORTC/MSG criteria of 2008 were most commonly used and there was only one study that used the newest update from 2020. The most common ODI cut-off value was 0.5. The antifungal prophylaxes used in the studies were fluconazole, itraconazole, caspogungin or l-AmB.

#### 3.2.2. BAL

In Appendix A, study characteristics of the included studies about BAL galactomannan are presented. In total, 12 studies were included with a total sample size of 1090 patients, including 241 patients with proven or probable IPA and 238 patients with possible IPA.

Additionally, in this case, all study populations included only adult (18+) patients with a hematological malignancy. The EORTC/MSG criteria of 2008 were most commonly used and there was no study which used the newest update from 2020. The most common cut-off value used was 1.0. The antifungal prophylaxes used in the included studies were fluconazole, itraconazole or l-AmB.

### 3.3. Methodological Quality of Studies Included in Meta-Analysis

The full assessment is presented in Appendix A. 

#### 3.3.1. Serum

In Appendix A, the methodological quality assessment and applicability per study included in the meta-analysis are shown. In the first domain, namely patient selection, five studies were graded to have a high risk of bias, mostly because they did not enroll patients consecutively/randomly or there was a case–control design. 

In the second domain, namely index test, no studies were graded to have a high risk of bias. In all studies, the cut-off used was pre-specified and all studies used the Platelia ELISA. It was often unclear whether the galactomannan assay was interpreted without knowledge of the reference standard. However, since there was a prespecified cut-off value, the interpretation could not be dependent on the reference standard and therefore this was not considered as a source of bias. 

In the third domain, namely reference standard, six studies were graded to have a high risk of bias, mostly due to incorporation bias. Four studies were graded to have an unclear bias since they did not specify whether there was an incorporation bias.

In the fourth domain, namely flow and timing, three studies were graded to have a high risk of bias because they did not include all selected patients in the final analysis.

The applicability concerns in domain one were considered to be high in only one study and to be unclear in two studies. In domains two and three, there were no applicability concerns in any study.

#### 3.3.2. BAL

In Appendix A, the methodological quality with the risk of bias assessment and the applicability per study included in the meta-analysis are shown. In the first domain, namely patient selection, four studies were graded to have a high risk of bias. This was mostly because they did not enroll patients consecutively/randomly or there was a case–control design. 

In the second domain, namely index test, two studies were graded to have a high risk of bias; one because they specified that not all investigators were blinded and one because the article did not specifically give an ODI cut-off value. All studies used the Platelia ELISA.

In the third domain, namely reference standard, three studies were graded to have a high risk of bias, mostly due to incorporation bias. One study was graded to have unclear bias since they did not specify whether there was incorporation bias.

In the fourth domain, namely flow and timing, five studies were graded to have a high risk of bias because they did not include all selected patients in the final analysis.

The applicability concerns were very low for the BAL GM studies. Only one study had unclear concerns in the patient selection domain. All the other studies had no applicability concerns.

### 3.4. Heterogeneity

The heterogeneity was first investigated by visual examination of the forest plots of all the data together. It is apparent that sensitivity and specificity vary largely across studies. Differences arise when considering different cut-offs, but even among the results from the same cut-off, there are non-overlapping confidence intervals, especially when looking at the sensitivity of the serum analysis. On the other hand, the reported specificity seems to be less variable across studies.

### 3.5. Primary Outcome

#### 3.5.1. Serum

In Appendix A, all the data per article are shown. The same data are plotted in forest plots (Appendix A). The analysis of all serum studies together, regardless of cut-off or study design, resulted in an overall sensitivity of 0.76 [95%CI, 0.60–0.87] and specificity of 0.92 [95%CI, 0.87–0.96] for proven/probable vs. no IPA (Appendix A). For proven/probable/possible vs. no IPA, an overall sensitivity of 0.45 [95%CI, 0.22–0.70] and specificity of 0.91 [95%CI, 0.86–0.95] was found (Appendix A). For the subanalysis for proven/probable vs. no-IPA excluding the serum studies with a population with a higher pre-test probability (i.e., studies that only included patients with fever), three studies were excluded [17,24,29]. This did not significantly alter the results (*p*-value = 0.21 for sensitivity and *p*-value = 0.42 for specificity). A similar subanalysis for the classification proven/probable/possible vs. no-IPA excluded three studies [17,28,39] and led to a similar conclusion (*p*-value = 0.53 for sensitivity and *p*-value = 0.31 for specificity). However, it must be noted that only three studies remained in the subanalysis.

In Appendix A, the pooled sensitivity and specificity per cut-off value are presented. For the ODIs 1.0 and 1.5, only one study was available per cut-off, which is insufficient for pooling and drawing any conclusions. For the group proven/probable IPA vs. no-IPA and the ODI 0.5, nine studies were available, which resulted in a pooled sensitivity of 0.92 [95%CI, 0.69–0.98] and a specificity of 0.84 [95%CI, 0.76–0.90]. Nine studies reported results regarding consecutive sampling with the ODI 0.5, which resulted in a pooled sensitivity of 0.66 [95%CI, 0.45–0.83] and a specificity of 0.95 [95%CI, 0.91–0.98]. For the analysis of proven, probable and possible IPA vs. no-IPA (Appendix A), five studies were available for the ODI 0.5, which resulted in a pooled sensitivity of 0.40 [95%CI, 0.05–0.89] and a specificity of 0.87 [95%CI, 0.71–0.95]. Additionally, five studies reported results regarding consecutive sampling with the ODI 0.5, which resulted in a pooled sensitivity of 0.55 [95%CI, 0.29–0.78] and a specificity of 0.94 [95%CI, 0.87–0.97].

#### 3.5.2. BAL

In Appendix A, we report all data found in the literature about BAL analysis. All data are plotted in forest plots (Appendix A). The analysis of all BAL studies together, regardless of cut-off or study design, resulted in an overall sensitivity of 0.80 [95%CI, 0.67–0.89] and a specificity of 0.95 [95%CI, 0.90–0.98] for proven/probable vs. no IPA. For proven/probable/possible vs. no IPA, it resulted in an overall sensitivity of 0.49 [95%CI, 0.25–0.73] and a specificity of 0.95 [95%CI, 0.87–0.98].

In Appendix A, the pooled sensitivity and specificity per cut-off value are presented. For the ODIs 0.8, 0.85 and 1.5, only one study was available, rendering these data unusable for pooling and drawing any conclusions. For the group proven and probable IPA vs. no-IPA and the ODI 0.5, six studies were available, which resulted in a pooled sensitivity of 0.75 [95%CI, 0.58–0.87] and a specificity of 0.88 [95%CI, 0.76–0.95]. For the ODI 1.0, eight studies were available, which resulted in a pooled sensitivity of 0.75 [95%CI, 0.56–0.88] and a specificity of 0.96 [95%CI, 0.91–0.98]. For the group proven, probable and possible IPA vs. no-IPA, three studies were available for the ODI 0.5, which resulted in a pooled sensitivity of 0.42 [95%CI, 0.73–0.99] and a specificity of 0.92 [95%CI, 0.19–0.99]. Additionally, three studies reported results regarding the ODI 1.0, which resulted in a pooled sensitivity of 0.62 [95%CI, 0.03–1.0] and a specificity of 0.99 [95%CI, 0.00–1.0].

## 4. Discussion and Conclusions

### 4.1. Summary of Main Results

In this study, we reviewed all the recent evidence regarding the use of both serum and BAL galactomannan in diagnosing invasive pulmonary aspergillosis. Defining the clinically relevant cut-offs is essential for clinical practice due to the high mortality rates in patients with a hematological malignancy. This is the first review addressing both serum and BAL GM in comparing all different hematological patient groups, including all IPA categories and performing clinically relevant subanalyses. All different cut-off values for the ODI for different IPA categories were analyzed, which resulted in an extensive overview of the current evidence. The results show that, for serum the ODI 0.5 and for BAL the ODI 1.0 portray the best diagnostic accuracy. 

Pooling the data for serum GM resulted in an overall sensitivity of 76% and a specificity of 92% for proven/probable vs. no IPA. The sensitivity dropped massively when considering the alternative categorization of proven/probable/possible, resulting in a sensitivity of 45% and a specificity of 91%. This result is reasonable because the likelihood of infection is fairly low for the possible IPA group. It is therefore relatively difficult to detect a true IPA in this patient group. 

For the BAL GM pooling, the data resulted in an overall sensitivity of 80% and a specificity of 95% for proven/probable vs. no IPA. Additionally, with BAL GM, adding the possible group resulted in a significant drop in sensitivity (49%); specificity remained similar (95%).

It has become clear that there is a lack of available research on this topic, especially for the rarer ODI cut-offs that are less commonly used. Our study affirms that the evidence for the use of GM in clinical practice for the hematological malignancy patient is currently insufficient and more research is needed to conduct the diagnostic value of GM.

### 4.2. Comparison to Other Reports

#### 4.2.1. Serum Galactomannan

There have been other reviews regarding the clinical use of serum galactomannan [9,11,40,41]. In our review, however, we included several recent studies which have not been used in a meta-analysis before and performed a more extensive subanalysis, making our review very relevant for clinical practice. In 2015, a study from the Cochrane Library by Leeflang reported a sensitivity and specificity for the cut-off ODI 0.5 of 0.79 and 0.80, respectively, for a single sample, and of 0.77 and 0.88 for, respectively, for subsequent samples [11]. For the cut-off ODI 1.0, they reported a sensitivity and specificity of 0.72 and 0.87, respectively, for a single sample, and of 0.70 and 0.92 for subsequent samples. Their patient group, however, is not fully comparable to our patient group: they also included children and other immunocompromised patients such as solid organ transplant recipients, patients with solid cancer and patients with HIV/AIDS. Additionally, they merged possible IPA together with the no-IPA group as negative controls. We decided not to include the possible IPA in the negative control group in our first analysis since these patients, in our opinion, cannot be regarded as truly not having IPA. This is in line with clinical practice wherein we often treat patients with possible IPA. This underlines the importance of our second analysis in which we include the possible IPA patient group. They also conclude that their numbers should be interpreted with caution because their results were very heterogeneous, which is in line with our own conclusion. 

Another publication describes a drop in the serum GM performance for non-neutropenic and non-hematological patients with a sensitivity range of 23.1–57.9 and a specificity range of 76.1–94.1 [42]. The reason for this is the availability of circulating neutrophils to clear the antigen. This affirms the clinical necessity of a systematic review with only neutropenic hematology patients. 

A 2006, review by Pfeiffer reported a pooled sensitivity of 0.61 and a specificity of 0.93 for all serum cut-off values [40]. Specifying it to different cut-off ODIs, they reported for cut-offs of 0.5 and 1.0 a sensitivity and specificity of 0.79 and 0.86 as well as 0.65 and 0.94, respectively. This review includes a variety of immunocompromised patients and they included the possible IPA’s as true negatives, but they did perform a subgroup analysis on only the hematological malignancy patient group which shows a sensitivity of 0.58 and specificity of 0.95 for all serum cut-off values. In this study they also conclude that the performance of the GM test drops sharply for solid-organ transplant recipients, reporting a sensitivity of 0.41 and specificity of 0.85. 

When comparing our serum results with the previous reviews, our diagnostic accuracy appears to be higher than previously found. This could be explained by differences in patient selection, e.g., using only neutropenic hematology patients as well as excluding the possible IPA patients from analysis. 

#### 4.2.2. BAL Galactomannan

Regarding BAL GM, a more recent review from Cochrane Library by De Heer was published in 2020 [9]. This review reports a sensitivity and a specificity of 0.84 and 0.83, respectively, at a cut-off ODI of 0.5 and a sensitivity and specificity of 0.80 and 0.92 for a cut-off ODI of 1.0. This review also included a variety of immunocompromised patients and no subgroup analysis was performed including only hematological patients. They also included 16+ patients, instead of 18+, and merged the possible label in the negative control (i.e., in the no-IPA group).

Another review by Heng from 2015 also analyzed the use of BAL GM, specifically focusing on an adult hematological patient population [41]. However, they also included the possible IPA patients with the no-IPA patients as negative controls. For the ODI 0.5, the sensitivity and specificity were 0.82 and 0.92, respectively, and for the ODI 1.0, the sensitivity and specificity were 0.75 and 0.95, respectively. 

Comparing our results to the aforementioned studies for both ODI 0.5 and 1.0, our results do not differ significantly. The slight differences may be explained by the differences in the patient groups.

### 4.3. Strengths and Weaknesses of This Review

Due to our strict in- and exclusion criteria, we ensured that the included articles were highly reliable and applicable on our study population of interest. The strength of our analysis lies in the multiple subgroup analysis we performed. By including only the no-IPA group as negative controls, we ensured that patients in this group most likely did not have IPA. Since we also conducted a separate analysis combining the possible IPA group with the proven and probable patient groups, we created an analysis which reflects diagnostic accuracy in clinical practice. This is also the first review to combine both serum and BAL galactomannan, which provides an extensive and highly reliable overview of the current evidence. 

Due to our strict criteria, we had to exclude some high-quality publications due to the fact that they included patients with other pre-specified significant diseases or when the age was 16+. This was also of influence on our final (smaller) sample size for the meta-analysis. Especially for the less reported ODI GM values (serum 1.0 and 1.5 and BAL 0.8 and 1.5), there were not enough data to pool and draw definitive conclusions. It is therefore not possible to endorse (nor wave aside) the use of a cut-off value of 1.0 for single serum galactomannan as advised by the EORTC/MSG 2020 guideline, based on our analysis. 

A potential source of bias that is worthwhile to discuss is the incorporation bias. As stated in the methodological quality assessment, the EORTC/MSG criteria implement the value of GM for the probable category, thus risking incorporation bias. 

Another potential source of bias is the time between the index test and the reference standard. Most studies are unclear about this and therefore we do not know its impact. It is however important to acknowledge this, since the positivity of the index test could be weeks before or after the time of diagnosis with the reference standard. Therefore, the true disease status of the patient could have been different at the time the index test turned positive. This could also explain the wide variety between the different study results. However, since the studies are not clear about this, the exact effect could not be estimated.

### 4.4. Conclusions

In conclusion, based on our data analysis, a serum ODI of 0.5 and a BAL ODI of 1.0 seem to be the most suitable cut-offs for clinical practice. Nonetheless, both have a mediocre sensitivity and specificity, which raises the question of whether GM is the right tool for detecting IPA early in its course and thus preventing the disease. Therefore, more research, preferably multiple randomized controlled trials, should be conducted.

## Data Availability

Upon reasonable request, data can be shared by contacting the corresponding author.

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
