# Peer review of "The Use of Galactomannan Antigen Assays for the Diagnosis of Invasive Pulmonary Aspergillosis in the Hematological Patient: A Systematic Review and Meta-Analysis"

_jof, 2023, doi:10.3390/jof9060674_

Round 1

Reviewer 1 Report

Bukkems et al have performed a systematic review and meta-analysis of the use of galactomannan (GM) for the diagnosis of invasive pulmonary aspergillosis (IPA).

The strict selection criteria for the population and study designs accepted make this paper unique in the literature and a valuable addition.

Points to address

1) Introduction, L 25:"...caused by Aspergillus species, commonly known as A. fumigatus." 

- This is NOT the only species of Aspergillus to cause IPA. Do you mean "the commonest cause of IPA"? Please clarify and correct

- Please italicise all pathogen names as per convention

2_ Methods, L69-70:"...entire population was currently using posaconazole prophylaxis were excluded ...". Please give a justification for this exclusion.

3) In the included studies, you have not commented on the use of antifungal prophylaxis. This is relevant to point (2) above, as you have specifically excluded studies on the basis of a specific drug and specific design (ie, all patients on that drug).

4) Results: it would aid the interpretation by clinicians if also presented the NPV/PPV values for at least the key cut-off GM values you recommend. 

5) Discussion, L377-378:"It is therefore apparent that GM is not the appropriate test for excluding IPA in this patient group." This warrants more thought - you say that the "the likelihood of the infection is fairly low for the possible IPA group". This means that the prevalence is low, hence the negative predictive value should be high in this setting - we would expect GM to be good at excluding disease as prevalence falls. 

6) You conclude in the Discussion (L384-386) that:"Our study affirms that the evidence for the use of GM in clinical practice for the hematological malignancy patient is currently insufficient and more research is needed to conduct the diagnostic value of GM. " This statement needs to be in the Summary as well - it is a key point that despite many years of clinical use and formal studies, the data of the utility of GM is lacking. 

7) It is worth commenting that the Leeflang et all paper also came to a similar conclusion - although they suggested optimal cut-offs for GM in clinical use, they concluded in the Summary:"These numbers should, however, be interpreted with caution because the results were very heterogeneous."

8) Point (6) above raises an important topic for the Discussion: given the lack of clear evidence on the utility GM for the diagnosis of IPA, how should the clinical diagnosis of GM be made?

9) L471-472:"Therefore more research should be conducted." This is not an acceptable final sentence! Please suggest what research is required; what is the optimal study design to get a definitive answer to this intractable question of the utility of GM in the diagnosis of IPA

Overall, no issues.

Reviewer 2 Report

a systematic review with meta-analysis to establish the  optimal cut-off value of the optical density index of the Galactomannan Antigen Assays (GM) for diagnosing invasive pulmonary aspergillosis in hematological patients.

The abstract needs to state the number of studies [n = 27, a good number of studies] that form the basis of the meta-analysis, the population of interest and the synthesis method [generalized linear mixed model with a binomial distribution]. Lines 15 – 16 it is unclear as to which results relates to serum, BAL, and at which ODI. The abstract needs a conclusion that aligns with lines 468-472.

Line 136 “This model has proven to be equivalent to the hierarchical summary ROC model.” Might be true – but unfortunately it fails to convey the inevitable trade-off between sensitivity and specificity over varying test thresholds and only a visual impression of heterogeneity. As a result your synthesis provides multiple and confusing summaries for sensitivity and specificity [within 21 figures and 19 tables] – whereas an SROC would provide a combined plot. This would be simpler. [Hurley J. Meta-analysis of clinical studies of diagnostic tests: developments in how the receiver operating characteristic “works”. Archives of pathology & laboratory medicine. 2011 Dec;135(12):1585-90.]

Line 62 “Any study design, except a review, was allowed” suggest “Any study design, except a review, was included for analysis”

Were any studies conducted in outbreaks?

Line 63 “…..no-IPA cases and controls were included.” Is unclear

Line 68-70 “Furthermore, publications where the entire population was currently using posaconazole prophylaxis were excluded and publications including the intensive care unit (ICU) population.” Is unclear – why exclude? Are intensive care unit (ICU) populations in or out and why? Does this limit generalizability? Presumably pediatric were excluded – correct? Why? This becomes a discussion point later.

Table 1 has 16 studies  - figure 1 indicate n = 15, one study [34] has two sub-studies but how do they differ – this is not clear. Is there any repeated measures?

Table 1 is a little difficult to read and could be simplified with strategic footnotes. What is ‘GM excluded by mycological criteria’? Is NA – Not available?

Table 2, one study [18] has two sub-studies but how do they differ – this is not clear. Is there any repeated measures?

Figures 6-onward - legends are unclear e.g. “Forest plot of diagnostic power of sensitivity serum galactomannan…” do you mean “Forest plot of sensitivity of serum galactomannan as a diagnostic assay for ….”

Some figures are poorly labelled – – what are the figure sub titles?

Major formatting errors which MUST be fixed - Tables are out of sequence and Table 16 is not visible.
